# Impacts of COVID-19 on Logistics Service Providers' Operations: An Italian Empirical Study

**Giovanni Zenezini** [ID], **Anna Corinna Cagliano** [ID], **Giulio Mangano** *[ID] and **Carlo Rafele** [ID]

Department of Management and Production Engineering, Politecnico di Torino, Corso Duca degli Abruzzi 24, 10129 Torino, Italy; giovanni.zenezini@polito.it (G.Z.); anna.cagliano@polito.it (A.C.C.); carlo.rafele@polito.it (C.R.)

* Correspondence: giulio.mangano@polito.it

**Abstract:** The lockdowns caused by the COVID-19 pandemic between 2020 and 2021 resulted in a substantial increase in e-commerce purchases, with the consequent growth of logistics services. Thus, this paper is aimed at studying the effects of the pandemic on the operational processes of logistics service providers. To this end, a survey questionnaire was developed and administered to a sample of identified respondents. The collected data were quantitatively analyzed via the Kruskal–Wallis test. The outcomes point out that logistics operators faced an increase in the distances traveled to carry out pick-up and delivery activities, and larger companies added more light vehicles to their fleets, proving that the company size was a relevant aspect of ensuring a quick response to the pandemic. These results show an increased business-to-consumer market share that is leading to a redesign toward more sustainable operational strategies.

**Keywords:** logistics service provider; COVID-19; operational sustainability; Italy; questionnaire

## 1. Introduction

The COVID-19 pandemic caused lockdowns, business closures, and other restrictions on the mobility of people and goods around the world in 2020–2021. These conditions led to a further increase in e-commerce purchases, with a consequent increase in the number of orders and associated deliveries, especially in the business-to-consumer (B2C) sector, where products and services are sold by companies directly to end users [1]. For example, Walmart reported a 97% increase in online sales in the second quarter of 2020 compared to the first quarter [2]. As a result, the demand for logistics services has increased dramatically, posing a major challenge to logistics service providers (LSPs), which have had to reorganize their operations in a sustainable way to adapt to market changes while dealing with issues such as quarantine periods, border closures, and the temporary unavailability of human resources.

Even before the pandemic, LSPs played a significant role in the global economy. For example, in 2018, they accounted for 8.5% and 10% of the GDP of the United States and the European Union, respectively [3]. During the most severe phases of COVID-19, LSPs became essential to ensuring that primary transportation services could meet the needs of both private end users forced to stay at home and vital business-to-business (B2B) sectors, such as food or pharmaceuticals. B2B refers to transactions where one company sells products and services to another company in the supply chain. To make the situation more complex, LSPs had to work in a dynamic environment during the many pandemic waves that occurred over time [4]. In fact, they had to deal with frequent and unexpected changes in the epidemiological situation and the resulting regulations to limit the spread of the virus. During the pandemic, erratic and volatile demand, government regulations, and supply chain disruptions led to financial failures, delivery delays, employee welfare

burdens resulting in employee unavailability, uncertainties in the forecasting horizon, increased home deliveries, increased product returns, and overall increases in transportation costs [5–7].

The impact of the aforementioned disruptions on the LSP business and the resulting responses have been the subject of a significant number of research studies. These works cover different topics: supply chain risk management [8,9], LSP resilience [4,10], economic and financial performance [8], how to ensure a sustainable competitive advantage [5,11], and innovation adoption [12]. The situations in different countries have also been analyzed: USA [2]; Thailand [13,14]; Malaysia, Indonesia, and India [15]; China [16]; South Africa [17]; UK [18]; Poland; and Germany and Sweden [12]. However, to the best of the authors' knowledge, very few studies have investigated the impact of COVID-19 on LSPs in Italy, despite the fact that it was one of the countries most affected by the pandemic in the world, especially during the first wave. Moreover, the existing literature focuses more on the strategic level regarding the financial and risk aspects of the COVID-19 pandemic and hardly considers operational processes. In fact, the increased level of stress in the operations of LSPs led to a greater risk of delayed deliveries [1] and increased transportation costs because of higher product returns [2]. In addition, COVID-19 acted as a barrier to innovation, with even LSPs with the right knowledge and resources deciding not to commit those resources due to the unpredictability of the COVID-19 pandemic aftermath [3]. To cope with the difficulties encountered during the pandemic, LSPs have resorted to a number of different responses, ranging from improving their communication and training their staff to introducing more interaction-less deliveries [4]. Nevertheless, the pandemic has created opportunities for LSPs, which may have been able to increase their revenue streams due to a lack of overall transportation capacity [5] in combination with an increase in online shopping [6]. In fact, ref. [7] Atayah, Dhiaf, Najaf, and Frederico (2021) found a positive impact of the pandemic on the financial performance of logistics companies in 14 of the G-20 countries. In addition, firms were driven to enhance and improve their ability to identify and respond to supply chain risks [8]. COVID-19 also proved to be an important driver of process digitization (Herold et al., 2021). However, the opportunities created by COVID-19 could only be exploited by LSPs with a high degree of dynamic resilience [9].

In order to fill the identified gaps in the literature, the present research was carried out to study the impact of the COVID-19 pandemic on the operational activities of LSPs operating in Northern Italy. The focus of this work is on this geographical area, as it was the first area in Europe to be put under lockdown. As a result, the LSPs operating in these areas were the first ones that had to reorganize their business activities. The aim of this paper is to assess how the main operational aspects, such as distances traveled, number of deliveries, order size, average order processing time, number of trucks, and staffing levels, changed in 2020, at the onset of the pandemic, compared to 2019. To better explain the variations in operating conditions, the study was conducted by examining pick-up and delivery costs, as well as some key strategic elements, such as investments in warehouse space and the implementation of new technologies. Possible differences related to the size of the company and the main market type, B2B or B2C, were taken into account. In addition, the point of view of different company professionals was taken into account. To this end, a survey was conducted among LSPs located in Northern Italy, and the data were analyzed using descriptive and inferential statistics. The main objective of the study is to help academics and practitioners better understand how LSP activities have been affected by the COVID-19 period. This could assist them in defining operationally sustainable strategies to survive in periods of highly unpredictable and radical market changes caused by exogenous factors.

This paper is structured as follows. First, an overview of the relevant literature is carried out. Then, the survey is presented, together with a description of the LSP operational aspects included. After that, the results are presented, and finally, the discussion and conclusions are provided.

## 2. Literature Review

The existing literature on COVID-19 and LSPs focuses only briefly on the operational impact that such a disruption may have caused. In other words, while several factors induced by the pandemic have been identified as having an impact on LSPs, only a few studies consider the magnitude of this impact. For example, ref. [10] measured the decline in international road transport for a sample of European countries by focusing on the organization of LSP processes. The research presented by [8] highlights that the respondents experienced major disruptions in their international supply chain networks due to the COVID-19 pandemic. These disruptions could be related to the capacity of ports, airports, and warehouses, as well as to the shortage of containers, and had a negative impact on the cost of freight transport. However, the effects of the COVID-19 pandemic spanned multiple domains, creating new revenue streams, increasing transportation flexibility, enforcing digitization, and optimizing logistics infrastructure and capacity [5].

In order to fill this gap, it is important to investigate which key operational variables and drivers of efficiency could be affected in the context of the more general impacts of COVID-19 mentioned above. For example, the reorganization of business processes may lead companies to look for new facilities to have more space available [11]. In addition, fleet size, which is one of the most important critical success factors for an LSP, should also be considered in terms of the vehicle mix (both small vans and trucks) [12], as it can become a source of tactical and operational advantages [13]. Finally, the impact on the size of the LSP's workforce, in terms of the number of drivers, warehouse operators, and office staff, should be considered as a proxy for the estimation of both resource availability and business volume [14].

The dramatic increase in e-commerce and B2C operations induced by the COVID-19 pandemic has had other effects. First, as a measure of the shift toward more B2C operations, the number of deliveries should be considered [19]. In addition, a lower weight-to-volume ratio of delivered items, as well as greater daily distances traveled by LSP fleet vehicles, can be included in the analysis as proxies for changes in the customer's B2C network [20]. Finally, another aspect that could have changed as a result of the increase in the B2C market share is the order size, which could be significantly lower [20]. Finally, the delivery time, i.e., the time from the receipt of the order to the final delivery to the customer, also plays a crucial role in terms of competitive advantage [16].

## 3. Research Methodology

### 3.1. Questionnaire Design

Based on the results of the literature review, a survey questionnaire was designed to understand the impact of COVID-19 on LSP operations. This topic is an evolving phenomenon, as LSPs are still facing the impact of the pandemic on their operations. Therefore, the survey method was chosen as an appropriate methodology to conduct this study [17]. In fact, surveys are effective for measuring unobservable data, such as the preferences and behaviors of firms [18], and they are easy to replicate even after a long period of time. In addition, surveys are inexpensive and allow for the remote collection of data on a population that is too large to be directly observed and for the detection of small effects even when analyzing multiple variables [21].

The questionnaire consists of two sections. The first contains relevant information useful to characterize the respondent companies, such as the market served by the company (i.e., B2B or B2C), the number of employees, and the professional role of the respondents, namely, Administrative Staff and Managers. According to the guidelines of the European Commission [22], companies with fewer than 10 employees were classified as micro-enterprises, those with 10–49 employees were classified as small enterprises, those with 50–250 employees were classified as medium enterprises, and those with more than 250 employees were classified as large enterprises. The second section includes specific questions assessing changes in logistics operations from 2019 to 2020 in terms of monthly deliveries, distances traveled, order lead times and sizes, fleet size, and employees. In

addition, the changes in the main performance indicators of logistics services in terms of timeliness and accuracy were examined. Other questions aim to assess the economic impact of the COVID-19 pandemic on LSPs in terms of the acquisition of warehouse space and the operating costs incurred in providing pick-up and delivery services. Finally, some questions have been dedicated to understanding whether technological solutions were implemented by LSPs in 2020 to better manage their transportation and warehousing operations and their impact on time and accuracy performance. In fact, the pandemic can potentially be a driver for the more extensive use of technologies to support logistics from both the provider and customer sides [22].

The survey questions were assessed by means of a 5-item Likert scale, where 1 = much decreased, 2 = slightly decreased, 3 = unchanged, 4 = slightly increased, and 5 = much increased.

Prior to its administration, the questionnaire underwent a pre-test with the aim of detecting possible criticalities related to the vocabulary, consistency, ambiguity, and redundancy of the questions. The questionnaire was also pre-tested to check for missing questions. For this purpose, expert judgment [23] was applied, and both academic and professional experts reviewed the draft questionnaire, which was then modified based on their suggestions. In addition, prior to the empirical quantitative analysis, the results obtained were tested using Cronbach's alpha coefficient to verify the internal consistency of the responses.

The complete questionnaire document is available in Appendix A.

### 3.2. Sample Selection

Three criteria were used to select potential respondents among the LSPs: (i) having at least one business unit located in Northern Italy, as it was the first European area to be strongly affected by the 2020 closures due to the COVID-19 infection; (ii) being characterized by a national activity code related to transportation and warehousing services; and (iii) having balance sheets available in the AIDA database (Analisi Informatizzata delle Aziende Italiane, https://login.bvdinfo.com/R0/AidaNeo, accessed on 18 February 2023), provided by Bureau Van Dijk, for the years 2019 and 2020. The AIDA database was chosen because it provides detailed and exhaustive quantitative financial data on companies operating in the Italian market. It is worth mentioning that the data available in the AIDA database at the time of this research, namely, 2022, refer to the financial situation of the companies until 2020. After an initial screening, a group of 400 LSPs was selected.

### 3.3. Questionnaire Administration

The questionnaire was developed using Google Forms. The administration period was two months, covering January and February 2022. An invitation to participate in the survey was sent first, followed by a reminder to those companies that had not yet completed the questionnaire about 20 days after the invitation. Responses were exported from Google Forms to an Excel spreadsheet, whose structure was then modified and adjusted to obtain a dataset suitable for subsequent statistical analyses.

### 3.4. Questionnaire Analysis Method

Descriptive statistics were first used to understand the general patterns of the data collected. Then, a quantitative statistical technique was chosen to delve deeper into the data analysis. Since the collected data are not normally distributed and the Likert scales used to rate the questionnaire responses are ordinal in nature, the Kruskal–Wallis nonparametric test was chosen to assess whether the medians of the associated populations were equal [24–26]. Kruskal–Wallis tests were performed here using the Minitab 17 software package. The test results were then interpreted.

The research methodology is summarized in Figure 1.

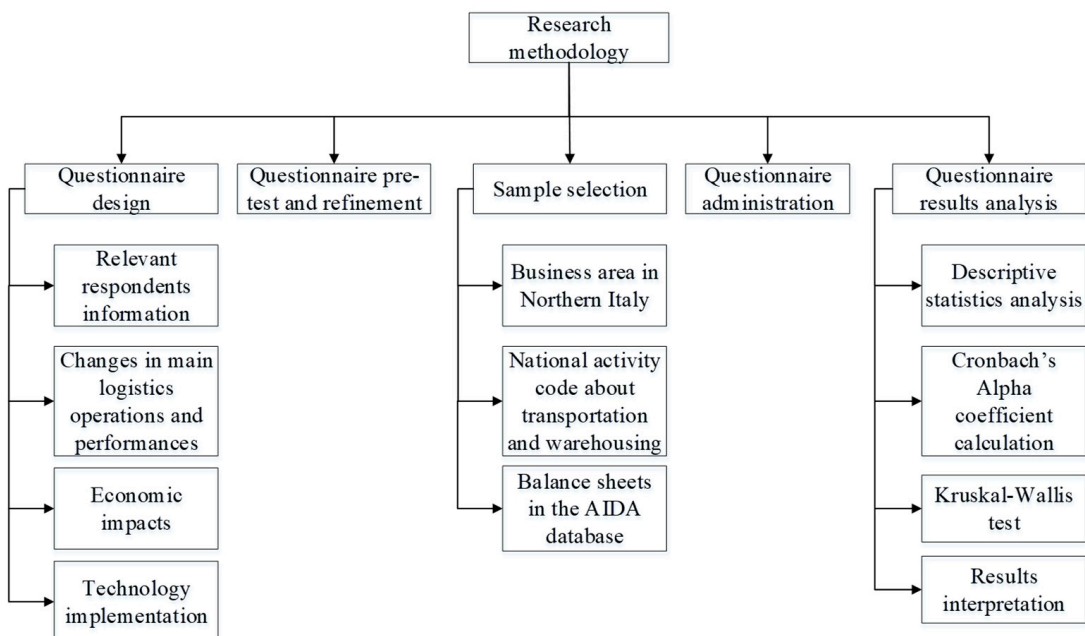

**Figure 1.** Research methodology chart.

## 4. Results

In the next sub-sections, key descriptive statistics are presented. Then, the Kruskal–Wallis test results are discussed and interpreted.

### 4.1. Sample Description

Of the 400 potential LSP respondents, 94 returned a complete questionnaire useful for subsequent analysis, giving a response rate of 23.5%. This is considered sufficient since online surveys have response rates between 14 and 22% [27].

Figure 2a shows the distribution of respondents according to their professional role: about two-thirds of respondents are managerial staff, who may have a better perception and outlook on business trends resulting from changes in production and distribution processes due to the pandemic.

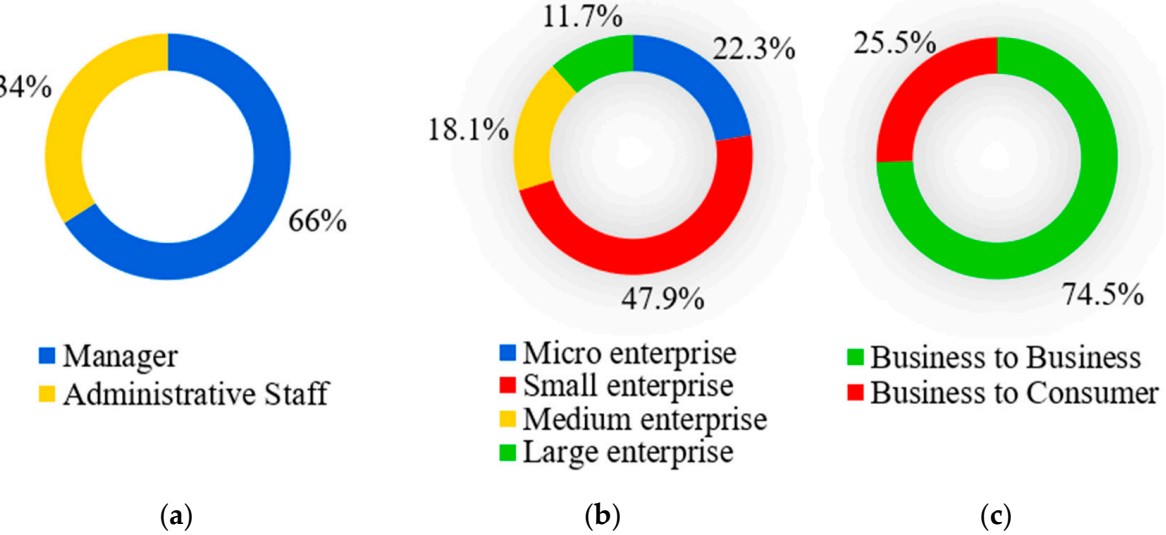

**Figure 2.** Distribution of respondents according to the professional role (**a**), company size (**b**), market type (**c**).

In terms of company size (Figure 2b), half of the respondents come from small companies, while the other half are almost evenly split between micro, medium, and large companies, showing that all LSP organizations, regardless of their size, have some interest in the topic. The larger number of small respondents is probably due to the fact that this is the dominant type of LSP operating in the geographical area studied. Finally, Figure 2c shows the distribution of respondents by market type and shows that about 75% of them operate mainly in a B2B environment. Such a result makes it possible to analyze the challenges faced by LSPs as a result of changing conditions not only in the purchasing habits of end users but also in industrial logistics practices.

### 4.2. Descriptive Statistics of Survey

The following sub-sections discuss the main evidence from the survey on LSP operations, investments in key resources, and the adoption of new technological solutions.

### 4.2.1. LSP Operations

The first result in line with expectations is the increase in the number of monthly deliveries in 2020 for 52% of respondents, with 33% of the LSPs surveyed reporting a very significant increase compared to 2019 values. The exponential growth of B2C e-commerce in the first half of 2020, as a result of retail store closures and travel restrictions due to the lockdown, is the main reason for this result. This trend was also confirmed for the second half of 2020, when partial lockdowns were imposed in selected geographical areas.

Some 41% of respondents reported a moderate-to-significant increase in order size, although 31% of companies said the average number of product units handled per order remained the same in 2020. Again, this is due to the surge in e-commerce during the early waves of the pandemic. In fact, more than half of LSPs operating primarily in a B2C environment experienced an increase in order size. The possibility of material shortages led end users to place larger orders than usual. Similarly, in the B2B sector, nearly 36% of respondents experienced an increase in the number of units per order. National and international transportation suffered from border closures, export bans, and good quarantine periods, which resulted in long lead times that caused companies to place larger orders with their suppliers less frequently [28].

While the size of orders has increased compared to the situation prior to the COVID-19 pandemic, the type of goods shipped has remained largely unchanged. In fact, 52 percent of those surveyed had no change in the ratio of the weight to the volume of shipments.

Figure 3 shows the changes in the average monthly distance traveled by the LSP fleet in 2020. In line with the growth in the number of deliveries, this has increased for around 37% of respondents.

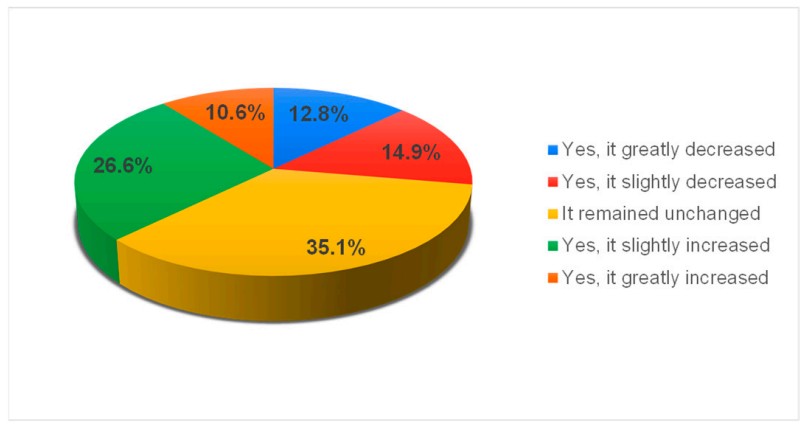

**Figure 3.** Changes in the average distance traveled.

For 51 percent of respondents, the average order lead time in 2020 was the same as in 2019, while 42 percent of respondents reported an increase, in some cases very significant.

These results show that LSPs have been able to maintain the same level of customer service during the early stages of the pandemic. This could be considered a good performance, as the time needed to perform logistics operations was affected by the combined effects of the increase in demand for services and the challenges of managing activities in the midst of the pandemic [29], together with the associated requirements, such as the adoption of social distancing rules. For example, LSPs were forced to either implement complete segregation between receiving and shipping areas or split activities across multiple shifts to reduce the number of operators working simultaneously in warehouses [30].

The overall reliability of LSPs, as measured by the number of correct and on-time deliveries, does not appear to have been affected by the COVID-19 pandemic. Correctness, measured as the percentage of error-free deliveries over the total number of deliveries, is equal to or greater than 95% for 80% of respondents, unchanged from 2019 values. The situation is similar for on-time deliveries, whose percentage is equal to or greater than 95% for 75% of respondents.

Finally, collection and delivery costs increased in 2020 for 74.5% of respondents, with 35% of them reporting a significant hike. Again, this could be due to both the growth in demand and the additional costs associated with complying with safety and social distance regulations.

### 4.2.2. Investments in Key Resources

The survey then analyses whether the corresponding increase in business volume as a result of closures and travel restrictions during the pandemic has led to new investments in key LSP resources such as vehicle fleets, staff, and warehouses.

The size of light commercial vehicle fleets, with a gross vehicle weight less than or equal to 3.5 tons, was unchanged for nearly 61% of respondents but was increasing for 32% of companies (Figure 4). Such a result highlights the tendency of LSPs to increase the size of their light van fleets, as these are the vehicles most used in urban environments, where the increase in the number of deliveries due to the pandemic was quite significant.

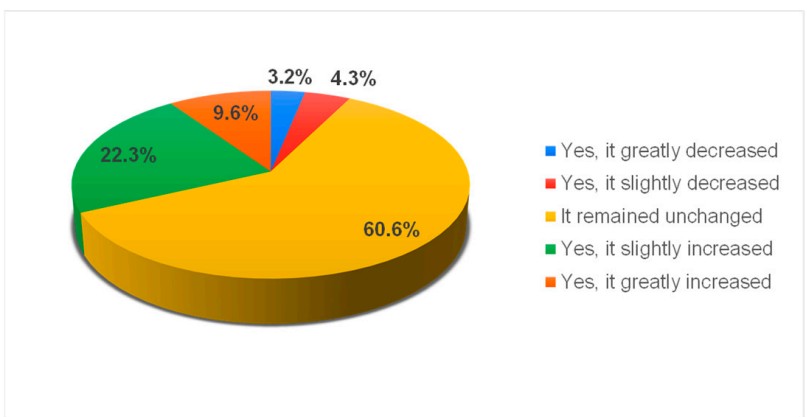

**Figure 4.** Changes in light commercial vehicle fleets.

Similarly, the survey results show an almost equal percentage of LSPs reporting unchanged (40.4%) and increased (43.7%) fleet sizes for commercial vehicles over 3.5 tons. Therefore, more companies chose to expand their heavy-truck fleets rather than light-truck fleets.

As expected, the number of employees involved in logistics operations increased for 51% of respondents in 2020 compared to 2019. The increased number of work shifts and quarantine periods that employees were subjected to required more staff, possibly temporary, to maintain the same level of customer service.

Finally, almost 44% of LSPs invested in the acquisition of storage space in 2020. As these tend to be medium- to long-term investments, these results show that a large

proportion of respondents see pandemic-related inventory growth as a long-term trend for the future.

### 4.2.3. Adoption of New Technological Solutions

In terms of technology adoption, half of the respondents said they had not implemented any new solutions. This may be due to the fact that the survey was conducted in early 2022, only two years after the pandemic outbreak, and technology decisions by companies often have a longer-term perspective. In other words, LSPs still need to clearly understand whether the market changes resulting from COVID-19 will be permanent and whether these trends will require the adoption of new tools. This is confirmed by the rather low percentage of LSPs adopting transportation management systems (TMSs) (17%) or warehouse management systems (WMSs) (18%) in 2020. In any case, LSPs focused on technological implementations related to order, warehouse, and transportation management, which is consistent with the need to achieve better control over orders, especially online orders, which increased significantly during the first wave of the pandemic. Overall, 38% of respondents said that technological solutions, when implemented, helped improve the accuracy and timeliness of deliveries. On the contrary, only 16% of companies benefited from reduced order lead times. As discussed in Section 4.2.1, the social distancing rules adopted in work environments increased service time.

### 4.3. Empirical Model

Before conducting the Kruskal–Wallis test, the Cronbach's alpha test was calculated for each question of the questionnaire to check the validity of the analysis (Table 1). This coefficient is a measure of internal consistency that assesses how reliable surveys are designed. It ranges between 0 and 1, with higher values indicating greater consistency [31]. Values greater than 0.7 can be considered sufficient to confirm the internal consistency and reliability of the scale [32].

**Table 1.** Values of Cronbach's alpha test.

| Item | Cronbach's Alpha |
|---|---|
| Number of deliveries | 0.8128 |
| Weight-to-volume ratio | 0.8218 |
| Distance traveled | 0.8096 |
| Delivery time | 0.8598 |
| Order size | 0.8181 |
| Changes in delivery without damages | 0.8382 |
| Change in on-time delivery | 0.8455 |
| Investment in storage area | 0.8202 |
| Cost of pick-up and delivery | 0.8518 |
| Fleet size less than 3.5 t | 0.8339 |
| Fleet size more than 3.5 t | 0.8296 |
| Total personnel | 0.8326 |

The results show that all coefficients are higher than 0.8, which means that the proposed questionnaire is properly designed to capture the perceptions of LSPs about the impact of the COVID-19 pandemic.

Table 2 shows the results of the Kruskal–Wallis test. The columns show the logistic items related to the questions administered to the respondents and the demographic aspects studied. For each, the median and the resulting *p*-value are presented. The significance level of the test was set at 5%. Significant relationships are highlighted in bond in Table 2.

**Table 2.** Kruskal–Wallis test results.

| Logistics Item | Business Position | | B2B/B2C | | Company Size | | | |
|---|---|---|---|---|---|---|---|---|
| | Staff | Manager | B2B | B2C | Micro | Small | Medium | Large |
| Number of deliveries | 3 | 4 | 3 | 4 | 4 | 3 | **4** | 5 |
| | | 0.475 | | 0.077 | | | **0.181** | |
| Weight-to-volume ratio | 3 | 3 | 3 | 3 | 3 | 3 | 3 | 3 |
| | | 0.914 | | 0.127 | | | 0.8 | |
| Distance traveled | 3 | 3 | **3** | **3.5** | 3 | 3 | 3 | 4 |
| | | 0.98 | | 0.033 | | | 0.413 | |
| Delivery time | 3 | 3 | 3 | 3 | 3 | 3 | 3 | 3 |
| | | 0.597 | | 0.781 | | | 0.291 | |
| Order size | 3.5 | 3 | 3 | 4 | 3 | 3 | 3 | 4 |
| | | 0.168 | | 0.09 | | | 0.148 | |
| Changes in delivery without damages | 3 | 3 | 3 | 3 | 3 | 3 | 3 | 3 |
| | | 0.711 | | 0.331 | | | 0.659 | |
| Change in on-time delivery | 3 | 3 | 3 | 3 | 3 | 3 | 3 | 3 |
| | | 0.656 | | 0.649 | | | 0.921 | |
| Investment in storage areas | 4 | 3 | 3 | 3 | 3 | 3 | 3 | 4 |
| | | 0.325 | | 0.231 | | | 0.276 | |
| Cost of pick-up and delivery | 4 | 4 | 4 | 4 | 4 | 4 | 4 | 4 |
| | | 0.251 | | 0.971 | | | 0.135 | |
| Fleet size less than 3.5 t | 3 | 3 | 3 | 3 | **3** | **3** | **3** | **4** |
| | | 0.3 | | 0.132 | | | **0.019** | |
| Fleet size more than 3.5 t | 3 | 3 | 3 | 3 | 3 | 4 | 4 | 3 |
| | | 0.721 | | 0.421 | | | 0.113 | |
| Total personnel | 4 | 3.5 | 4 | 3.5 | 3 | 4 | 3 | 5 |
| | | 0.983 | | 0.535 | | | 0.057 | |

The results show that the total distance traveled increased more for companies operating in the B2C market than for companies operating in the B2B environment. This may depend on the pandemic-induced e-commerce boom dramatically increasing B2C activity, which will affect the distances that LSPs have to cover to fulfill orders. Such a finding is consistent with the increased number of vans with a gross weight of less than 3.5 tons that can be observed in the fleets of large companies.

## 5. Discussion

The present work contributes to the state of the art on the impact of COVID-19 on LSPs by providing an operational perspective focused on the Italian situation.

The overall picture that emerges from this research is that LSPs operating in Italy faced a significant increase in the number of deliveries, the size of orders, and, consequently, the average distance traveled following the COVID-19 pandemic outbreak and the associated social restrictions and lockdowns imposed to try to contain the spread of the virus. In addition, the pandemic has had a lasting impact on customers' lifestyles, and it is therefore highly likely that the impact on LSPs will continue after the pandemic is over. In general, such a situation could compromise customer service levels if not managed appropriately. These findings are in line with the general awareness of the impact of COVID-19 on LSPs

that emerges from the literature, confirming that Italian LSPs have followed the same path as their counterparts in other countries.

In addition, the LSPs in the study demonstrated their ability to keep pace with changing demand and business conditions by investing in new commercial vehicles, new warehouse space, and additional staff. These efforts to maintain overall operational performance levels came at a cost. In fact, the average order lead time promised to customers increased for many LSPs, as did the economic burden associated with pick-up and delivery activities, compared to the period immediately preceding the pandemic. In the future, a more extensive network of logistics facilities, especially in urban areas (e.g., parcel lockers, urban hubs), should be used to reduce transport distances and, more generally, negative externalities [33].

Finally, LSPs seem reluctant to adopt new technological solutions, waiting to see how the logistics business stabilizes after the pandemic before investing in IT solutions. Furthermore, LSPs have not invested in new technologies even though they have experienced an increase in order lead times, which may indicate that the order lead times currently offered by LSPs are considered adequate by consumers. However, few companies have tried to adapt to the disruptions and increases in volume by using solutions such as WMSs or TMSs, which aim to organize and coordinate order fulfillment and transportation [34]. These solutions are already consolidated in the industry, and thus, the pandemic may have driven the digitalization of LSPs that are either the late majority or even laggards in their technology adoption. Such a result may be due to the fact that in times of increased revenues, companies are less willing to invest, as they are able to make a profit without seeking help in specific investments, which are often highly customized and cannot be easily redeployed to alternative value-creating uses [35]. Thus, it is reasonable to expect that in the near future, with the return to a stable level of competition, firms will look for new levers of competitive advantage. In this context, technology adoption could be considered as a suitable option.

This empirical study provides additional insights into the impact of the pandemic on different categories of LSPs. In fact, the increase in the number of kilometers traveled for pick-up and delivery services is mainly due to the increased market share of the B2C sector, which is driven by e-commerce. This dramatic change has led companies operating in the B2C context to reorganize their operational processes. The reported increase in lead times for fulfilling orders received and higher logistics costs may be due to the fact that companies were not prepared to face an unexpected situation and, in turn, were not able to pursue a relevant level of efficiency. The company size also plays a role in the response to the pandemic. Large companies were found to have added more light vehicles to their fleets in order to easily respond to rapid changes in volume, a finding that could be explained by the greater availability of financial resources to such companies. In addition, large companies may be able to find available drivers more easily, especially at a time when there is a shortage of van drivers [8]. Thus, large firms can be considered to be better able to deal with the uncertainties associated with the fleet issue [1]. Furthermore, according to the survey results, the proportion of large companies operating in the B2C market that have responded to the business pressure exerted by COVID-19 by purchasing light commercial vehicles is significantly higher than that of LSPs in the B2B market. As a result, we may be witnessing a trend toward greater vertical integration in the last-mile delivery of e-commerce goods, as large companies make the decision to purchase new vehicles themselves and move away from outsourcing the last-mile transport of goods to local companies. As a result, the transport and storage capacities of the new LSPs may be greater than the volume of new business.

## 6. Conclusions

The present research examines the responses of LSPs operating in the Northern Italian market to the COVID-19 pandemic. It explores how these providers have adapted their operations to meet the new requirements and to cope with the disruptions caused by

this situation. A survey was distributed to a cohort of 400 companies, including micro, small, medium, and large enterprises, selected from those with established records in a well-known financial database. Ninety-four complete responses were received, and the data were analyzed using both descriptive and inferential statistical methods. The results indicate that LSPs have experienced an expansion of their business scope, primarily opting to increase their resources, such as vehicles, warehouses, and labor. However, these efforts have come at a cost, resulting in longer order lead times and economic burdens associated with pick-up and delivery activities. Interestingly, LSPs have been cautious in adopting new technological solutions, perhaps waiting for business to stabilize after the pandemic before considering investments in IT solutions. While some companies have implemented WMSs or TMSs to manage disruptions and volume increases, the industry as a whole appears to be hesitant to adopt new technologies.

This study also highlights differences in responses between LSP categories, with increased distances traveled linked to the growing market share of the B2C e-commerce sector. Large companies showed greater adaptability than smaller ones, as evidenced by the addition of light vehicles to their fleets, possibly due to their greater financial resources and flexibility in responding to volume changes. These findings provide insights that can help LSPs develop operationally sustainable strategies to maintain their competitive position in the face of unexpected and disruptive events caused by external factors.

There are a few limitations to this study. The first is related to the time horizon of the study, which does not include any period beyond 2020. Second, the geographical focus is limited. In fact, the results represent only the Italian situation, with particular attention to the northern, more industrialized areas of the country, and as such, they cannot be generalized at either the European or global level. Such limitations provide insights for future research efforts. Extending the geographical scope of the empirical research developed would allow the results of the present work to be validated. To this end, it would be beneficial to carry out the survey at the European level in order to understand whether the results related to LSPs operating in Northern Italy are in line with the international logistics trends associated with COVID-19. At least the largest European countries should be included in the survey in order to be able to make some comparisons between different national environments. Increasing the scope and geographical coverage of the study may require the use of advanced data analysis techniques, such as machine learning and deep learning. In addition, the results of this work may raise a question for readers. What happens now that all social and trade restrictions due to COVID-19 have been removed? For example, further research could address both the customer demand and the operational environment faced by LSPs in the current era of transition to an endemic stage of COVID-19 [36]. Therefore, future studies could investigate how the operational environment of LSPs, as well as their business models, has been structurally changed by COVID-19. In this context, it would be interesting to understand whether the current transport and storage capacities of LSPs appear to be oversized compared to the business volume from the pandemic period. Regarding the strategies of LSPs, it would be interesting to know whether and how the increase in business caused by COVID-19 could act as an incentive for LSPs to improve their level of digitization in the medium and long term.

In addition, the present work could encourage future research on multimodal transportation solutions as a potential cost-effective alternative when distances traveled increase, as happened during the pandemic period. Finally, in order to have a multivariate perspective that includes more economic aspects, the market share and income of each company studied could also be included in the analysis and related to the size of the company.

**Author Contributions:** Methodology, G.M.; Investigation, G.Z.; Writing—original draft, A.C.C.; Writing—review & editing, C.R. All authors have read and agreed to the published version of the manuscript.

**Funding:** This research received no external funding.

**Institutional Review Board Statement:** Not applicable.

**Informed Consent Statement:** Informed consent was obtained from all subjects involved in the study.

**Data Availability Statement:** Data about the present research work will be shared upon request.

**Conflicts of Interest:** The authors do not have any relevant financial or non-financial competing interests.

## Appendix A. Administered Survey

| |
|---|
| 1. You work as: Manager/Administrative Employee |
| 2. Your most important market is: Business to Consumers/Business to Business |
| 3. The Company you work for is: Small/Medium/Large |
| 4. Has there been any changes in the total number of deliveries per month from after the pandemic? (1 they have decreased a lot–5 they have increase a lot) |
| 5. Has there been any changes in weight to volume ratio transported per month by the vehicle fleet from after the pandemic? (This is defined as the ratio of the weight of products to the volume they occupy) (1 it has decreased a lot–5 it has increased a lot) |
| 6. Has there been any changes in the distances traveled in a month by the vehicle fleet after the pandemic? (1 they have decreased a lot–5 they have increase a lot) |
| 7. Has there been any change in the average time elapsed from the time a shipment request is received until it is received by the client (including loading/unloading transports) after the pandemic? (1 it has decreased a lot–5 it has increased a lot) |
| 8. Was there a change in order size, defined as the average number of units for which customer handling is required, after the pandemic? (1 it has decreased a lot–5 it has increased a lot) |
| 9. By considering the percentage of deliveries without damages, after the pandemic its value is: 1 it has decreased a lot 5 it has increased a lot |
| 10. By considering the percentage of order delivered on time, after the pandemic its value is: 1 it has decreased a lot 5 it has increased a lot |
| 11. Has there been any change in the investments associated with the increased of the storage areas in terms of purchasing or leasing of new logistics facilities after the pandemic, after the pandemic? (1 they have decreased a lot–5 they have increase a lot) |
| 12. Have there been any changes in the costs for picking and delivery processes goods after the pandemic? (1 they have decreased a lot–5 they have increase a lot) |
| 13. Has there been any change in the number of vehicles with a capacity of less than or equal to 3.5 tons after the pandemic (1 it has decreased a lot–5 it has increased a lot) |
| 14. Has there been any change in the number of vehicles with a capacity greater than 3.5 tons, after the pandemic (1 it has decreased a lot–5 it has increased a lot) |
| 15. Has there been any change in staffing levels after the pandemic? (1 it has decreased a lot–5 it has increased a lot) |

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
