# Peer review of "Impacts of COVID-19 on Logistics Service Providers’ Operations: An Italian Empirical Study"

_sustainability, doi:10.3390/su16010208_

Round 1

Reviewer 1 Report

Comments and Suggestions for Authors

This manuscript may be reconsidered after major revisions.

1. L24: Full name and detailed information of B2C should be provided.

2. Reference [3] did not provide enough information as required, making it difficult to be searched. In fact, these data vary significantly with years, and the manuscript did not specify which year the data was from.

3. L35: Full name and detailed information of B2B should be provided.

4. L37-40: Rewrite this sentence.

5. There are many unreasonable paragraph settings in the Introduction section. Some of them are suggested to be merged.

6. L79-97: This paragraph seems more suitable for moving to the Introduction section.

7. Generally believed, the Literature Review section should provide a good overview of the similar works in the field. In the manuscript, significant improvements are needed.

8. L125: This sentence can be deleted.

9. L156-157: “owning or renting an operational facility located in Northern Italy It does not mean that the main business area of the samples is in Northern Italy, so the impact of the restriction will be disrupted.

10. Fig.1: What is the basis for dividing the company size into micro, small, medium, and large enterprises?

11. L220-239: Relevant data details should be provided as figures or tables, at least in the supplementary materials.

12. Fig. 2 and 3: Pie charts seem more suitable.

13. L385: The Conclusion section is not appropriate and requires some modifications, as many of its contents, especially those related to limitations and future insights, are more suitable for moving to the Discussion section.

Comments on the Quality of English Language

Extensive editing of English language required

Author Response

This manuscript may be reconsidered after major revisions.

  1. L24: Full name and detailed information of B2C should be provided.

R: The full name has been reported for the acronym B2C and a short definition of this notion has been added.

  1. Reference [3] did not provide enough information as required, making it difficult to be searched. In fact, these data vary significantly with years, and the manuscript did not specify which year the data was from.

R: In Section 1 of the manuscript we stated the year the data reported in line 33 refer to, which is 2018. Additionally, in the Reference section we added the link to the pdf file of the report associated with reference [3].  

  1. L35: Full name and detailed information of B2B should be provided.

R: The full name has been reported for the acronym B2B and a short definition of this notion has been added.

  1. L37-40: Rewrite this sentence.

R: This sentence has been rewritten as follows in order to improve its readability:

“To make the situation complex, LSPs had to work in a dynamic environment during the many pandemic waves that occurred over time [4]. In fact, they had to face frequent and unexpected changes of the epidemiological situation and consequent regulations to limit the virus spread”.

  1. There are many unreasonable paragraph settings in the Introduction section. Some of them are suggested to be merged.

R: The paragraph setting in Section 1 has been revised and some of the paragraphs in the original version of the manuscript have been merged. Please check the revised version of Section 1.

  1. L79-97: This paragraph seems more suitable for moving to the Introduction section.

R: The afore mentioned paragraph has been moved to the Introduction Section

  1. Generally believed, the Literature Review section should provide a good overview of the similar works in the field. In the manuscript, significant improvements are needed.

The Literature Review has been extended by adding several papers considering how LSP have been faced the COVID pandemic as shown below:

R: “In other words, the potential impacts of the pandemic on the service level have been explored only to determine which factors did have an impact on LSPs rather than the magnitude of that impact that is taken into account only on a few studies. For instance [6], measure the drop in terms of international traffic of road carriers for a sample of European countries By focusing on the organization of LSPs’ processes the research proposed [7] by highlights that the respondents had experienced major disruption across its international SC network because of the COVID-19 pandemic. These disruptions could be referred to capacity and transportation infrastructure of ports, airports and warehouses and to container shortages with negative effects on freight costs. However, the effects of COVID pandemic have included different domains namely, create new revenue streams, enhance the transport flexibility, enforce digitalization, and optimize logistics infrastructure and capacity [8].”

  1. L125: This sentence can be deleted.

R: The mentioned sentence has been deleted.

  1. L156-157: “owning or renting an operational facility located in Northern Italy” It does not mean that the main business area of the samples is in Northern Italy, so the impact of the restriction will be disrupted.

R: Thank you for the comment. Indeed, this criterion for sample selection was not formulated correctly in the manuscript to reflect its real meaning. It has been now changed as:

“ having at least one business area located in Northern Italy”

  1. Fig.1: What is the basis for dividing the company size into micro, small, medium, and large enterprises?

R: The basis for dividing the company size into micro, small, medium, and large enterprises has been now explained in Section 3.1 Questionnaire Construction, where the following sentence has been added:

“In particular, according to the European Commission guidelines [33], the companies with less than 10 employees were classified as micro enterprises, those with 10-49 employees as small enterprises, those with 50-250 employees as medium enterprises, and those with more than 250 employees as large ones.”

  1. L220-239: Relevant data details should be provided as figures or tables, at least in the supplementary materials.

R: The relevant data details related to what discussed in the mentioned lines of Section 4.2.1 of the paper have been provided in form of pie charts as supplementary material for review.

  1. Fig. 2 and 3: Pie charts seem more suitable.

R: Figure 2 and Figure 3 have now been presented as pie charts.

  1. L385: The Conclusion section is not appropriate and requires some modifications, as many of its contents, especially those related to limitations and future insights, are more suitable for moving to the Discussion section.

R: Dear reviewer, thank you for the comment. We restructured the conclusion section to include a summary of the paper and thus we believe that the conclusion section is now more appropriate. To answer to your comment on the limitations and future insights however we believe they belong to the conclusion section. Here we wanted to highlight that there two major limitations of our study namely the geographical scope and the time horizon of the survey. These limitations, which confine the scope of this present study, may call for future research which are also introduced in the conclusion section.

Nevertheless, we have moved part of the conclusion section to the discussion section, namely the sentences regarding the changed demand pattern that appears to be systemic after the Covid-19 pandemic as shown below:

Moreover, given the enduring impact of the pandemic on customer lifestyles, it is highly unlikely that the demand faced by LSPs will revert to pre-pandemic levels. Simultaneously, the existing demand likely differs from that observed during the different phases of the pandemic.”

Comments on the Quality of English Language

Extensive editing of English language required

R: A thorough revision of the English language used in the paper has been performed.

Reviewer 2 Report

Comments and Suggestions for Authors

The paper investigates the effect of the COVID-19 pandemic on the operations processes of Logistics Service Providers (LSPs) in the north of Italy. The study used a survey questionnaire to collect data from identified respondents.

The article needs be revised before it can be published. Before presenting the results, the authors should further enhance their research methodology based on the use of the questionnaire, such as sample calculations to ensure representative results, validity and reliability testing, and so on.

Author Response

The paper investigates the effect of the COVID-19 pandemic on the operations processes of Logistics Service Providers (LSPs) in the north of Italy. The study used a survey questionnaire to collect data from identified respondents.

The article needs be revised before it can be published. Before presenting the results, the authors should further enhance their research methodology based on the use of the questionnaire, such as sample calculations to ensure representative results, validity and reliability testing, and so on.

R: In Section 3.1 the Authors have explained the reasons why a survey questionnaire was identified as Methodology. Am appropriate reference has been inserted coherently as shown below

“This topic is a developing phenomenon because LSPs have been faced the effects of the COVID pandemic on their operations. Thus, the survey method was selected as appropriate methodology for carrying out this study. In fact, surveys are effective for measuring unobservable data such as companies preferences and behavior. Furthermore, surveys are economical and allow for remotely collecting data about a population that is too large to observe directly and detect minor effects even while analyzing multiple variables [15]”.

The reliability of data has been tested by means of the Cronbach’s Alpha coefficient as mentioned in Section 4.3. However, this aspect has been underlined in Section 3.1 too as shown below.

“In addition, before undertaking the empirical quantitative analysis, the obtained results were tested via the Cronbach’s Alpha coefficient in order to check the internal consistency of the responses.”

Reviewer 3 Report

Comments and Suggestions for Authors

The article is of significant interest, as it is based on an empirical study that allows us to assess the impact of the Covid-19 pandemic on the activities of LSPs. However, in my opinion, the authors should not have limited themselves to studying the situation in Northern Italy, but considered, at least briefly, the situation in other European countries.

Author Response

The article is of significant interest, as it is based on an empirical study that allows us to assess the impact of the Covid-19 pandemic on the activities of LSPs. However, in my opinion, the authors should not have limited themselves to studying the situation in Northern Italy, but considered, at least briefly, the situation in other European countries.

R: Thanks for your comment. The authors understand the need to extend the geographical scope of the study to other European countries. This has been pointed out in Section 6, first as a limitation of the research presented in the paper:

“the outcomes depict the Italian situation only, with particular attention to the Northern, more industrialized, areas of the country, and, as such, they cannot be generalized at a either European or worldwide level”

and then as a future research direction:

“it would be beneficial to perform the survey at a European level, in order to understand whether the outcomes related to LSPs operating in Northern Italy are consistent with the international logistics trends determined by Covid-19. At least the largest European countries should be involved in the survey, in order to be able to draw some comparisons among different national environments. In such a context, it would be also interesting to analyze any possible relationships between the pandemic trends and the LSP operations in each country.”

Reviewer 4 Report

Comments and Suggestions for Authors

Dear Authors,a very appreciated work which background's has a social relevance,having been the supply chain the only opportunity of contact with the real and external world,during the pandemia.

Inside your paper:

-The questionnaire in two sections(the 1st as a classification of LSP's operators,the 2nd as comparative) is perfectly adequate.

The choice of territorial base for the study,demonstrates to be highly significant.

However,the percentage of response is below the normal expectations,and could be increased on follow up procedures.

The dynamics of market share related to the size of companies,can be a further proposal on a multivariate focus.

On the considered criterias:

-Comparative distances are adequately integrated;

-Cost's increases focus is satisfactory expressed;

-Comparative investment's analysis is adequately considered;

-Comparative innovation is well considered.

-As an invitation for a future reasoning,a comparison of income sorted by company's sizes might be considered.

The Methodology adopted by proposal of the Cronbach Alpha Test as a measure of reliability as a first step,then on Kruskal Wallis non parametric Test(on the median values) is appropriated.

Further,your study shows clearly the permeability between the two main types of Logistics(B2B respectively B2C).

Some consequent main questions might bring a reason to continue this approach.

1)Out of pandemia time,were the LSP's oversized?

2)Did the companies consider the business increase as the incentive to increase the level of digitalisation?

3)As you have demonstrated that the distances increased,would have a multi-modal solution(train+last mile on truck) brought an alternative,together with a cost reduction?

Maybe this is not realistic on small business sizes,but could be considered by main LSP's as a challenge for the future,together with a change of mentality.

Author Response

Dear Authors,a very appreciated work which background's has a social relevance,having been the supply chain the only opportunity of contact with the real and external world,during the pandemia.

Inside your paper:

-The questionnaire in two sections(the 1st as a classification of LSP's operators,the 2nd as comparative) is perfectly adequate.

R: Thank you for your comment.

The choice of territorial base for the study,demonstrates to be highly significant.

However,the percentage of response is below the normal expectations,and could be increased on follow up procedures.

R: The administration of the survey was completed by already a follow up procedure, consisting   carrying out an additional wave in order to maximize the response rate. This value at end of the procedure was equal to 23.5% that is considered an acceptable value for further analysis. This outcome, was supported by appropriate literature (see Monzon et al. 2020). In absolute terms, the final sample was equal to 94 respondents that is a value that allows empirical measurements.

The dynamics of market share related to the size of companies, can be a further proposal on a multivariate focus.

R: Thanks. The authors have introduced this topic as a future research direction in Section 6:

“Finally, in order to take a multivariate perspective including more economic aspects, the market share and the income of each investigated company could also be considered in the analysis and related to the company size.”

On the considered criterias:

-Comparative distances are adequately integrated;

-Cost's increases focus is satisfactory expressed;

-Comparative investment's analysis is adequately considered;

-Comparative innovation is well considered.

-As an invitation for a future reasoning, a comparison of income sorted by company's sizes might be considered.

R: The authors have also introduced this topic as a future research direction. Please see the previous answer.  

The Methodology adopted by proposal of the Cronbach Alpha Test as a measure of reliability as a first step,then on Kruskal Wallis non parametric Test(on the median values) is appropriated.

Further,your study shows clearly the permeability between the two main types of Logistics(B2B respectively B2C).

Some consequent main questions might bring a reason to continue this approach.

1)Out of pandemia time,were the LSP's oversized?

2)Did the companies consider the business increase as the incentive to increase the level of digitalisation?

3) As you have demonstrated that the distances increased, would have a multi-modal solution (train+last mile on truck) brought an alternative, together with a cost reduction?

Maybe this is not realistic on small business sizes, but could be considered by main LSP's as a challenge for the future, together with a change of mentality.

R: The three comments above have been an appreciated opportunity to enlarge the discussion on future research directions in Section 6 as follows:

“In this regard, it would be interesting to understand whether the current LSP transport and storage capacity appear to be oversized compared to the business volume out of the pandemic time. About LSPs strategies, a topic that is worth exploring concerns whether and how the business increase caused by Covid-19 might act as an incentive for LSPs to improve their level of digitalization in the medium and long run. Additionally, the present work might foster future research on multimodal transportation solutions as a potential cost-effective alternative when distances travelled increase as it happened in the pandemic period.”

Reviewer 5 Report

Comments and Suggestions for Authors

This study analyzes the impacts of Covid-19 on Logistics Service Providers' operations—an Italian empirical study. While the topic is interesting, conducted during Covid-19, and holds potential for future studies, my main concern lies with the methodology and structure of the current methodology. The methodology relies solely on descriptive statistics, which might not be sufficient for a prominent SCI journal. It is too simple and naïve.  The authors should consider employing state-of-the-art methodologies to contribute significantly to the science and domain of transportation and logistics. While the contextual contribution from an Italian perspective is understandable, the study should incorporate a more comprehensive methodology suitable for such journals.

If the dataset permits, utilizing machine learning and deep learning techniques could enhance the study's scope and readability. Additionally, it's crucial for the authors to include the questionnaire in the appendix for transparency. The abstract should be refined following the IMRAD rule, providing a global perspective on the study rather than focusing solely on Italy or a specific country. Research of this nature should aim for universality.

In Section 3.1, consider renaming it to "Questionnaire Design" and provide details on how many questions were removed during the pilot stages and the reasons for their exclusion. Adding a flowchart in the methodology section can illustrate the step-by-step process, enhancing clarity. Here are a few resources that could be helpful for the authors in structuring the questionnaire administration and understanding the steps involved in similar papers:

These resources can provide insights into organizing the questionnaire, determining sample size, conducting reliability tests, pilot surveys, and handling the removal of responses. Incorporating these aspects into your research can enhance its completeness and methodological rigor. https://www.mdpi.com/2071-1050/14/5/2778‎, https://www.hindawi.com/journals/jat/2021/5141798/. 

If Section 3.4 pertains to results analysis, it should be moved to the results section. In Section 4, choose either "Results" or "Analysis" for consistency. Address confusing headings like "Sample Description"; consider consolidating data and methods or keeping descriptive analysis in the results section.

In the conclusion, reorganize the content to first present the objectives, followed by the applied methodology and data, and then the main findings. Provide a summary of the paper before discussing limitations and recommendations.

Comments on the Quality of English Language

The introduction paragraph should be revised coherently and properly structured. Please check for acronyms and typos throughout the entire manuscript. Especially, ensure that the section headings' terms are appropriate.

Author Response

This study analyzes the impacts of Covid-19 on Logistics Service Providers' operations—an Italian empirical study. While the topic is interesting, conducted during Covid-19, and holds potential for future studies, my main concern lies with the methodology and structure of the current methodology. The methodology relies solely on descriptive statistics, which might not be sufficient for a prominent SCI journal. It is too simple and naïve.  The authors should consider employing state-of-the-art methodologies to contribute significantly to the science and domain of transportation and logistics. While the contextual contribution from an Italian perspective is understandable, the study should incorporate a more comprehensive methodology suitable for such journals.

R: The proposed analysis deals with both qualitative and qualitative approaches. In particular, the data obtained via the survey have been presented at a qualitative level in Sections 4.1 and 4.2 (namely Sample Description and Descriptive Statistics of the Survey). On the contrary, in Section 4.3 the empirical model is presented. In particular, the data have been analyzed with a quantitative perspective by means of the Kruskal- Wallis test that allows to investigate differences in terms of opinions among different group of respondents on specific topic.

If the dataset permits, utilizing machine learning and deep learning techniques could enhance the study's scope and readability.

R: Thanks for this suggestion. The dataset has been designed for the application of statistical tools, such as the Kruskal-Wallis test one, and in its current form is scarcely suitable to the application of Machine Learning and Deep Learning. However, by enlarging the dataset size and when analyzing more aspects of the observed phenomenon, together with their mutual relationships, Machine and Deep Learning might become useful. So, the authors will consider their application in the next steps of the research. This was acknowledged in the Section 6 of the manuscript, as part of the discussion of the future research directions. In particular, the author think the adoption of these advanced data analysis techniques might be beneficial when extending the empirical research to cover more operational and strategical aspects of LSPs business in multiple European countries.

Additionally, it's crucial for the authors to include the questionnaire in the appendix for transparency. The abstract should be refined following the IMRAD rule, providing a global perspective on the study rather than focusing solely on Italy or a specific country. Research of this nature should aim for universality.

R: The Abstract has been revised according to the IMRAD rule, by discarding the information related to Italy, and including the main discussion aspects as shown below

These results show an increased Business to Consumers market share that is bringing to a rede-sign toward more sustainable operational strategies.

In Section 3.1, consider renaming it to "Questionnaire Design" and provide details on how many questions were removed during the pilot stages and the reasons for their exclusion.

R: In that phase questions associated with the performances have been added. In addition, in the initial version of the survey, the question referring to the vehicle fleet did not consider the vehicle size. Thus, it has been split in two questions in order to assess the impact of both small and large vehicles.

 Adding a flowchart in the methodology section can illustrate the step-by-step process, enhancing clarity. Here are a few resources that could be helpful for the authors in structuring the questionnaire administration and understanding the steps involved in similar papers:

These resources can provide insights into organizing the questionnaire, determining sample size, conducting reliability tests, pilot surveys, and handling the removal of responses. Incorporating these aspects into your research can enhance its completeness and methodological rigor. https://www.mdpi.com/2071-1050/14/5/2778‎,  https://www.hindawi.com/journals/jat/2021/5141798/. 

R: A flow chart of the methodology has been added as Figure 1.

The suggested references have been added in Section 3.1 for better describing the reasons why the questionnaire survey has been selected as appropriate methodology

If Section 3.4 pertains to results analysis, it should be moved to the results section. In Section 4, choose either "Results" or "Analysis" for consistency. Address confusing headings like "Sample Description"; consider consolidating data and methods or keeping descriptive analysis in the results section.

R: Section 4 has been renamed consistently 

In the conclusion, reorganize the content to first present the objectives, followed by the applied methodology and data, and then the main findings. Provide a summary of the paper before discussing limitations and recommendations.

R: The conclusion section has been reorganized following the reviewer’s comments and now presents a summary of the objectives, methods and main findings of the paper. 

Comments on the Quality of English Language

The introduction paragraph should be revised coherently and properly structured. Please check for acronyms and typos throughout the entire manuscript. Especially, ensure that the section headings' terms are appropriate.

R: The Introduction Section has been reshaped by moving a paragraph from the Literature Review, and by adding a sentence for more accurately explaining the aim of the research

Round 2

Reviewer 1 Report

Comments and Suggestions for Authors

Most of the responses to the comments were roughly satisfactory. The manuscript could be accepted for publication.

Comments on the Quality of English Language

Minor editing of English language required

Author Response

Thank you

Reviewer 2 Report

Comments and Suggestions for Authors

The article can be published in present form.

Author Response

Thank you

Reviewer 5 Report

Comments and Suggestions for Authors

The authors have adequately revised the manuscript as I requested. Kindly change Section 3.4 to 'Questionnaire Analysis Method' instead of 'Result.' The term 'result' is not appropriate.Kindly check all the references, I have found problem in it. thanks

Comments on the Quality of English Language

Minor check of English is required.

Author Response

Section 3.4 has been renamed.

The reference list has been updated accordingly